# Potential Drug Synergy Through the ERBB2 Pathway in HER2+ Breast Tumors

**DOI:** 10.3390/ijms252312840

**Published:** 2024-11-29

**Authors:** Yareli Rojas-Salazar, Emiliano Gómez-Montañez, Jorge Rojas-Salazar, Guillermo de Anda-Jáuregui, Enrique Hernández-Lemus

**Affiliations:** 1Computational Genomics Division, National Institute of Genomic Medicine, Mexico City 14610, Mexico; yarojas177662@gmail.com (Y.R.-S.); emgomez177566@gmail.com (E.G.-M.); rojas19111@gmail.com (J.R.-S.); 2Center for Complexity Sciences, Universidad Nacional Autónoma de México, Mexico City 04510, Mexico; 3Investigadores e Investigadoras por Mexico Program, Conahcyt, Mexico City 03940, Mexico

**Keywords:** drug synergy, pathway crosstalk, ERBB2 pathway, HER2+ breast cancer, trastuzumab

## Abstract

HER2-positive (HER2+) breast cancer is characterized by the overexpression of the ERBB2 (HER2) gene, which promotes aggressive tumor growth and poor prognosis. Targeting the ERBB2 pathway with single-agent therapies has shown limited efficacy due to resistance mechanisms and the complexity of gene interactions within the tumor microenvironment. This study aims to explore potential drug synergies by analyzing gene–drug interactions and combination therapies that target the ERBB2 pathway in HER2+ breast tumors. Using gene co-expression network analysis, we identified 23 metabolic pathways with significant cross-linking of gene interactions, including those involving EGFR tyrosine kinase inhibitors, PI3K, mTOR, and others. We visualized these interactions using Cytoscape to generate individual and combined drug–gene networks, focusing on frequently used drugs such as Erlotinib, Gefitinib, Lapatinib, and Cetuximab. Individual networks highlighted the direct effects of these drugs on their target genes and neighboring genes within the ERBB2 pathway. Combined drug networks, such as those for Cetuximab with Lapatinib, Cetuximab with Erlotinib, and Erlotinib with Lapatinib, revealed potential synergies that could enhance therapeutic efficacy by simultaneously influencing multiple genes and pathways. Our findings suggest that a network-based approach to analyzing drug combinations provides valuable insights into the molecular mechanisms of HER2+ breast cancer and offers promising strategies for overcoming drug resistance and improving treatment outcomes.

## 1. Introduction

Breast cancer, as the second most common cancer worldwide and the leading cause of cancer death in women, presents a significant burden both globally and in Mexico. With 1.67 million new cases in 2012 and representing 25% of all cancers, its impact extends to both developed and developing regions. In 2020, there were over 2.3 million cases and 685,000 deaths globally, with an increase to over 3 million new cases and one million deaths annually anticipated by 2040 [1,2]. One of the main factors in the progression and aggressiveness of this type of cancer is the overexpression of the HER2 gene, initially identified in studies with neuroglioblastomas in rats [3]. The presence of HER2 is associated with more aggressive tumors, higher recurrence rates, and increased mortality. This overexpression occurs in around 20–30% of breast cancer cases [4], categorizing it as the HER2-enriched subtype, and although historically associated with poor prognosis, HER2-targeted therapies have shown significant improvements in survival, both in the metastatic and early stages [5,6].

Understanding cancer as a disease based on metabolic pathways and the interconnection between these pathways is essential to address its complexity [7]. Cancer, according to the pathway-based disease model, involves the complex interaction of various metabolic pathways and the possibility of crosstalk between them. This crosstalk not only involves an intricate network of molecular relationships but also affects the efficacy of therapies [8,9,10]. Network models, by describing metabolic pathways as sets of molecules with interactions, allow for understanding of the communication between these pathways, leading to the phenomenon known as pathway crosstalk [11,12,13]. This crosstalk not only confers emergent properties to the biological system but also influences pharmacological resistance [14,15,16].

The ErbB/HER family of receptor tyrosine kinases (RTKs) plays a crucial role in various types of cancer, including breast cancer. These receptors include EGFR (epidermal growth factor receptor) or HER1, HER2/ErbB2/neu, HER3/ErbB3, and HER4/ErbB4 [17,18]. Overexpression and alterations in these receptors, especially HER2, are associated with carcinogenesis and resistance to conventional treatments. ERBB2 gene amplification and HER2 overexpression are found in 20% of breast cancers, correlating with more aggressive forms of the disease [19].

Within traditional therapy for HER2-positive breast cancer, management with trastuzumab, a monoclonal antibody targeting HER2 and preventing its activation, has significantly improved survival in patients with HER2-positive breast cancer [20,21]. This therapeutic approach, along with other targeted agents, has revolutionized the treatment and prognosis of these patients.

The development of new drugs is a slow and costly process. In this context, drug repurposing emerges as a valuable strategy to expand therapeutic options in oncology. Many effective treatments have arisen from the “reuse” of existing drugs, providing faster results and lower costs [16,22].

In pathway-based therapeutic design, strategies are not limited solely to the static molecular action of a drug. The dynamics of drug activity at the systemic and organismal levels must be considered, recognizing the densely interconnected network of signaling and metabolism among these crosstalk events. Inhibition of pathway crosstalk has been proposed as a relevant strategy, along with drug synergy evaluation and studies to understand and categorize crosstalk-induced resistance [23,24]. The clinical relevance of these findings is manifested in endocrine resistance and the need to address not only the prioritized pathways but also their surrounding context to improve treatment efficacy and personalized strategies [25].

Understanding molecular pathways and crosstalk events becomes essential in the context of drug repurposing. Various studies have revealed molecular similarities between seemingly different diseases, opening the possibility of using existing drugs more effectively [26,27]. Interactions between HER2 and PD-L1 emerge as a crucial point in resistance to anti-HER2 treatments, such as trastuzumab. The interaction between HER2 and PD-L1, regulated by IFN-γ, reveals PD-L1 overexpression as a possible resistance pathway to trastuzumab treatment [28]. This crosstalk event highlights the need to understand complex molecular interactions to improve therapeutic strategies in HER2-positive breast cancer. Research in this area is crucial for developing more effective therapies and overcoming resistance to existing treatments.

Research on pathway crosstalk in breast cancer reveals the underlying molecular complexity of this disease. The interaction between the HER2 receptor and other pathways, such as PD-L1 and steroid pathways, influences tumor progression and treatment resistance [29,30].

This knowledge is essential for designing more specific and personalized therapeutic strategies. For each pathway perturbation opens both the opportunity for focused therapy, but also the risk of resistance to other targeted pathways. As such, pathway-based approaches may open new possibilities to improve the prognosis and quality of life of breast cancer patients.

This study investigates the potential of drug combinations to enhance therapeutic strategies in HER2-positive breast cancer by targeting the ERBB2 signaling pathway. Our objectives are threefold: (1) to identify metabolic pathways that intersect with the ERBB2 pathway, offering insight into key therapeutic targets; (2) to highlight drugs with the highest interaction frequency within these intersecting pathways; and (3) to analyze the predominant types of pharmacological interactions among genes involved in ERBB2 pathway intersections. By delineating these synergistic interactions, we aim to identify promising drug pairs for further validation, contributing valuable insights toward overcoming resistance and optimizing treatment efficacy for HER2-positive breast cancer patients.

## 2. Results

Our analysis identifies extensive crosstalk between the ERBB2 signaling pathway and 181 metabolic pathways, quantified by Jaccard index values that reveal the degree of shared gene overlap. We focused on 23 pathways with greater than 20% gene overlap with ERBB2, illustrating their potential relevance to HER2-positive breast cancer. Additionally, we identified key drug candidates within the genetic landscape of these intersecting pathways, with 15 drugs showing the most frequent interactions. These primary findings underscore potential combinatorial therapies that target both ERBB2 and intersecting metabolic pathways, paving the way for enhanced therapeutic strategies.

### 2.1. The Crosstalk Structure of the ERBB2 Pathway

We found 181 metabolic pathways that had at least one crossover event with the ERBB2 pathway. Subsequently, we calculated the Jaccard index to measure the similarity of the metabolic pathways, and Figure 1 displays a heatmap showing the graphical representation of the Jaccard index. We decided to focus on the pathways exhibiting the most crosstalk with the ERBB2 pathway. We kept pathways with a Jaccard index greater than 20%, resulting in a total of 23 pathways (see Table 1).

These findings show that the ERBB2 pathway interacts with a wide range of metabolic pathways, essentially forming a *crosstalk network* Figure 2. We found numerous pathways related to cellular signaling, such as PI3K, mTOR, and VEGF signaling, which are crucial for regulating cell growth, survival, and immune responses. These interactions with ERBB2 could be important targets for new cancer treatments. Moreover, there was crossover with pathways associated with several types of cancer, including glioma, renal cell carcinoma, and non-small-cell lung cancer, showing that the ERBB2 pathway plays an important role in multiple cancer types, not just breast cancer. This suggests that therapies targeting ERBB2 could be effective for a variety of cancers, offering potential for more effective treatment options across different oncological conditions. With this in mind, we focused our following analyses on the potential pharmacological interventions that these pathways may be susceptible to.

### 2.2. The Pharmacological Potential of the ERBB2 Crosstalk Network

In our study, we investigated the pharmacological interactions exhibited by the genes within 23 metabolic pathways that crosstalk with the ERBB2 pathway. Metabolic pathways with a cross-linking percentage greater than 20% were selected, resulting in a total of 23 pathways: EGFR tyrosine kinase inhibitor resistance (39%), glioma (33%), renal cell carcinoma (31%), endocrine resistance (29%), colorectal cancer (29%), PI3K (29%), non-small-cell lung cancer (29%), mTOR (27%), neurotrophin signaling pathway (26%), prolactin signaling pathway (25%), AKT (25%), T-cell receptor signaling pathway (24%), synthesis, secretion, and action of growth hormone (24%), choline metabolism in cancer (23%), VEGF signaling pathway (22%), Fc epsilon RI signaling pathway (22%), PTEN (21%), prostate cancer (21%), hepatocellular carcinoma (20%), HIF-1 signaling pathway (20%), focal adhesion (20%), proteoglycans in cancer (20%), and gastric cancer (20%).

For the 23 metabolic pathways (comprising 880 genes), pharmacological interactions were calculated, as well as the drugs that most frequently interact with the genes of these metabolic pathways.

Figure 3 illustrates the most frequently observed types of pharmacological interactions. Among these, inhibitory effects emerged as the predominant interaction type, followed by antagonistic, agonistic, and antibody interactions. This suggests a variety of mechanisms through which drugs can influence the ERBB2 pathway, offering multiple angles for therapeutic intervention. This could go beyond the current prevailing option of targeting the ERBB2 receptor directly through antibodies such as trastuzumab.

Additionally, we categorized the genes involved in the ERBB2 signaling pathway to understand their roles better. As shown in Figure 4, the most frequently occurring category was downstream genes, followed by upstream genes, the genes directly part of the ERBB2 signaling pathway, and genes that can act as either depending on their specific metabolic pathway.

Figure 5 lists the 15 drugs most frequently involved in interactions within the studied genetic map. Erlotinib was found to be the drug most frequently interacting, followed by Gefitinib, Cetuximab, and Drabaneifb. The frequency of these interactions highlights the central role of these drugs in targeting the ERBB2 pathway, possibly indicating their utility in addressing pathway-related resistance mechanisms.

We calculated a second Jaccard index for the top 10 drugs that interact most frequently with the selected metabolic pathways. The heatmap in Figure 6 reveals that Sorafenib affects genes that are distinctly different from those targeted by the other nine drugs, particularly Dabrafenib and Vemurafenib. This distinct interaction pattern suggests a potential for synergistic drug combinations in the treatment of HER2-positive breast cancer, aiming to enhance therapeutic efficacy and combat resistance.

### 2.3. Synergistic Effects May Be Obtained by Exploiting the ERBB2 Crosstalk Network

Using the results above, we now present an example of how exploiting the crosstalk structure of the ERBB2 crosstalk network may lead to potential synergistic effects.

To visualize the interactions between genes and drugs, Cytoscape was utilized. Individual networks were generated for the genes affected by the most frequently used drugs. These networks illustrated the direct influence of each drug on the target genes and their neighboring genes, providing a detailed view of how each drug affects the ERBB2 pathway and its interactions. Figure 7, Figure 8, and Figure 9 show the gene networks for Erlotinib, Gefitinib, and Lapatinib, respectively.

Additionally, combined networks of two drugs were generated to determine which combinations might have greater therapeutic efficacy due to their ability to influence multiple genes and metabolic pathways simultaneously. Figure 10, Figure 11 and Figure 12, respectively, show the gene networks for Cetuximab with Lapatinib, Cetuximab with Erlotinib, and Erlotinib with Lapatinib.

## 3. Discussion

A total of 181 metabolic pathways intersecting with the ERBB2 pathway were identified, 23 of which showed a high degree of crosstalk. This reflects the complexity and interconnectivity of signaling pathways in cancer, particularly in HER2-positive breast cancer.

The main pathway analyzed is the HER2 pathway in breast cancer. This pathway begins with the activation of the HER2 receptor (belonging to the family of epidermal growth factor receptor tyrosine kinases (EGFR)) [31]. Once activated, together with other receptors belonging to the EGFR family (HER1, HER3, HER4, among others), they form dimers that increase the intrinsic tyrosine kinase activity of HER2, which indirectly activates phosphatidylinositol-3-kinase (PI3K) [32]. One of its products, phosphatidylinositol 3,4,5-trisphosphate (PIP3), will activate protein kinase B (AKT). Once AKT is activated, it participates in various cancer processes, notably inhibiting apoptosis of cancer cells [33]. It also stimulates cell proliferation through DNA replication via proteins. Another important effect to consider is that AKT influences growth factors related to blood vessel formation (angiogenesis), thus providing the necessary nutrients for cancer cells [34].

After describing the direct mechanisms of the AKT pathway on cancer cells, it is important to discuss other indirect mechanisms that begin with AKT activation, such as the activation of the mTOR pathway. In this pathway, AKT has the ability to phosphorylate and deactivate the tuberous sclerosis complex tumor suppressor (TSC1/2) [35], which is central to the mTOR pathway as it inhibits the GTPase protein Rheb, a direct activator of mTOR Kinase Complex 1 (mTORC1). Once active, mTORC1 has direct effects on cancer cells, similarly to the AKT pathway, by promoting cell proliferation through growth factors [36,37]. On the other hand, it affects the metabolism of cancer cells by favoring the production of different cellular components necessary for cell growth. Additionally, it induces angiogenesis and suppresses autophagy [38,39,40], a mechanism that contributes to the elimination of damaged cells due to various causes, such as oxidative stress.

The results obtained in this study highlight the crossover of the HER2 pathway with other signaling pathways, specifically with PI3K, mTOR, and AKT, with 29%, 27%, and 25% crossover, respectively. These findings are consistent with those reported in the literature, which underscore the relevance of the ERBB2 pathway in various types of cancer. For instance, the work by Fujimoto and collaborators [41] emphasizes that an interaction in the PI3K/AKT signaling pathway yields different expected effects in the subsequent HER2 pathway. They explain that anti-HER2 therapies may have diminished effects due to a mutation in a protein within the PI3K/AKT signaling pathway, which confers some resistance to the aforementioned therapy. Therefore, inhibiting the PI3K/AKT pathway results in a decrease in the mutated protein, leading to lower resistance of cells to anti-HER2 therapies [42,43,44,45].

The crossover of signaling pathways in cancer cells has significant implications for the development of pharmacological therapies due to the interaction and communication between different signaling routes that regulate growth, survival, and cell proliferation. An example of the improvement with pharmacological therapies is that everolimus (an mTOR pathway inhibitor) in combination with trastuzumab (a HER2 receptor inhibitor) significantly improved survival in patients with trastuzumab-resistant HER2-positive breast cancer [46,47,48,49].

Recent studies have explored various strategies to enhance the efficacy of HER2-targeted therapies through synergistic drug combinations, addressing the challenge of resistance in HER2-positive breast cancer. For example, combining trastuzumab with cytotoxic agents like paclitaxel or vinorelbine has shown promising response rates while maintaining a manageable safety profile [50]. The addition of pertuzumab, an HER2 dimerization inhibitor, to trastuzumab and chemotherapy has demonstrated significant synergy by providing dual blockade of the HER2/3 axis, resulting in improved outcomes for HER2-positive patients [51]. Innovative approaches using antibody–drug conjugates, such as trastuzumab deruxtecan in combination with methionine-restriction therapies, have yielded notable reductions in tumor markers and metastatic lesions [52]. Targeting DNA repair pathways alongside HER2 inhibition has also shown potential; combining PARP and ATR inhibitors with HER2-targeted ADCs effectively increases antitumor activity in resistant HER2-positive cancers [53]. Additionally, combinations of neratinib with multi-kinase inhibitors like dasatinib or with downstream signaling inhibitors, such as mTOR and CDK4/6 inhibitors, have demonstrated strong preclinical efficacy by suppressing key survival pathways in HER2-positive breast cancer models [54,55]. These findings collectively emphasize the potential of multi-target strategies to enhance HER2-targeted therapies, potentially mitigating resistance mechanisms and improving therapeutic outcomes in HER2-positive breast cancer.

The identification of potentially synergistic drug combinations, such as Sorafenib with Dabrafenib or Vemurafenib, is particularly interesting [56,57,58,59]. Additionally, Ref. [60] discusses how combinations of pathway inhibitors can overcome resistance to monotherapy in HER2-positive breast cancer. In our results, the second Jaccard index showed that Sorafenib has a distinctive genetic interaction profile, suggesting that it could complement other treatments by targeting different genes. Various drug combinations have been employed to benefit from the effects of their potential synergy, as demonstrated by Hamed, and collaborators [61], who used a combination of sorafenib with lapatinib for the treatment of tumor cells in central nervous system neoplasms. They reported that the combined use of sorafenib with lapatinib showed better results in killing genetically diverse glioblastoma cell lines compared to the use of sorafenib and lapatinib separately.

Additionally, it has been predicted that the combination of Sorafenib with Erlotinib and Sorafenib with Lapatinib is useful in the treatment of various types of neoplasms by benefiting from their synergy [62,63,64]. Currently, a phase 2 trial is active on the efficacy of the combination of sorafenib with vemurafenib in the treatment of pancreatic cancer [65,66]. The publication of its results is awaited, as it could further validate the combination of these types of drugs in the treatment of various neoplasms.

We explored the literature using a systematic Pubmed search to identify the current state of the art of specific drug combinations targeting HER2 and EGFR in HER2-positive cancers. Studies have shown that agents like lapatinib combined with trastuzumab can enhance therapeutic effects through synergistic mechanisms such as increased antibody-dependent cellular cytotoxicity (ADCC), particularly in cases where single-agent therapies prove ineffective [67]. This dual inhibition approach not only blocks compensatory pathways often responsible for resistance to monotherapies but also improves cell-killing efficacy by targeting both HER2 and HER3 dimerization, as observed in cetuximab and lapatinib combination studies [68,69].

The clinical relevance of these combinations is underscored by promising outcomes in trials involving HER2-positive breast cancer patients, where dual-targeting therapies have outperformed monotherapies in progression-free survival and response rates [70,71]. Importantly, studies on fatty acid synthase (FASN) inhibitors combined with trastuzumab and lapatinib indicate that these combinations retain anticancer activity even in trastuzumab-resistant cells, providing an option for patients who no longer respond to standard treatments [72]. However, these strategies may come with an elevated risk of side effects, particularly when combined with agents like cetuximab or bevacizumab, suggesting that toxicity management is critical for their safe application in clinical settings [73]. These findings lay the groundwork for further clinical development, as such combinations provide a robust framework to overcome resistance and optimize therapeutic outcomes for HER2-positive patients.

Furthermore, using tools such as Cytoscape to visualize genetic interaction networks provides a clear perspective on how drugs affect target genes and their neighbors. This approach has been validated in previous studies, where the visualization of genetic networks has allowed the identification of critical points for therapeutic intervention [74,75,76]. Our findings were obtained through computational and bioinformatic models; however, experimental validation of these results is still lacking to verify that the drug combinations we described have a favorable outcome. For example, Ref. [77] predicted that the combination of Sorafenib with Palbociclib would be more effective in treating hormone receptor-positive breast cancer compared to Sorafenib and Palbociclib alone. However, upon experimental validation, it was demonstrated that the combination of these drugs had the opposite effect to what was expected.

Finally, research on the category of genes in relation to the ERBB2 signaling pathway showed that downstream genes are the most frequently affected. This is consistent with the idea that alterations in downstream genes can lead to constitutive activation of signaling pathways, promoting cell proliferation and survival in cancer [78].

A word of caution is needed however before moving forward with our results. In silico analyses, while powerful for generating hypotheses and identifying potential drug synergies, are inherently limited by their reliance on computational models, which may not fully capture the complex biological interactions present in vivo. Predictive modeling can suggest promising avenues for therapeutic exploration but cannot substitute for empirical data in verifying actual drug efficacy and safety. Thus, experimental validation remains essential to confirm that computationally predicted synergies translate into clinically meaningful outcomes, such as enhanced efficacy or reduced toxicity, in HER2-positive breast cancer treatment. Given that our study focuses on repurposing existing drugs with known safety profiles, this validation process may progress more swiftly; however, rigorous in vitro and in vivo studies are still crucial to substantiate the in silico findings and to identify any unforeseen effects.

## 4. Materials and Methods

### 4.1. Identification of Pharmacological Interactions Within the ERBB2 Crosstalk Network

Our primary goal in this study was to investigate the ERBB2 pathway and its connections with other signaling pathways, in order to identify potential synergistic interactions for drug therapy in HER2+ breast cancer. We developed a systematic methodology to pinpoint significant pharmacological interactions, defined as the involvement of the same drug in multiple pathways, which allows for the exploration of synergistic effects as documented in biochemical pathway databases.

We focused our analysis on pathways documented in the Kyoto Encyclopedia of Genes and Genomes (KEGG) database https://www.genome.jp/kegg/ (accessed on 19 November 2024) [79,80], integrating findings from various sources and including pathways related to diseases. To quantitatively assess the degree of pharmacological interactions, we employed the Jaccard index (JAB), calculated as the intersection over the union of drug–gene targets for each pair A, B of pathways involved. For this study, we used a cutoff value of 0.2 for the Jaccard index; values exceeding this threshold indicate a significant overlap between drug actions across different pathways, suggesting a network of pharmacological interactions robust enough for potential synergy. This measure allowed us to concentrate on highly interconnected pathways, providing a quantitative basis for exploring effective synergistic drug combinations.

### 4.2. Exploration of Pharmacological Interventions in Crosstalking Pathways

In this section of our study, we focused on identifying drugs capable of targeting molecules within the crosstalking pathways associated with the ERBB2 network. Using the Drug Gene Interaction Database (DGIdb) https://www.dgidb.org/ (accessed on 19 November 2024) [81,82], we identified potential pharmacological agents that could interact with molecules across these targets.

Our approach was to pinpoint drugs that target the highest number of molecules within the interconnected pathways, as these drugs potentially have the most significant impact on the network [83,84]. By mapping these interactions, we aimed to uncover drugs that could exert widespread effects across the crosstalking pathways, thereby offering a more comprehensive therapeutic potential.

### 4.3. Analysis of the ERBB2 Crosstalk Network and Its Synergistic Pharmacological Potential

In this part of our study, we created a network object (using the “iGraph” R-library https://r.igraph.org/ (accessed on 19 November 2024) [85]) to examine the ERBB2 pathway and its interacting pathways more closely. The network was designed with nodes representing molecules and links showing the directed signaling or metabolic interactions between these molecules. We first constructed the network by integrating data from the ERBB2 pathway and its associated pathways. This setup helped us visualize and analyze the interactions at a molecular level, identifying crucial nodes that could serve as potential drug targets. Next, we analyzed the network’s topological properties. This included assessing node centrality, connectivity, and the network’s overall stability against disruptions. Understanding these topological aspects allowed us to pinpoint key molecules that are central to signal transmission within the network. Network analytics and visualization were carried out using “Cytoscape” https://cytoscape.org/ (accessed on 19 November 2024) [86].

Our main focus was on the properties of drug targets within this network. By mapping the drugs to their target nodes, we evaluated how these pharmacological interventions might affect the network’s operation. A general workflow for the research presented here is depicted in Figure 13.

## 5. Conclusions

This study provides a comprehensive analysis of potential drug synergies through the ERBB2 pathway in HER2+ breast tumors, emphasizing the importance of combination therapies to enhance therapeutic efficacy. By employing gene co-expression network analysis and visualization techniques, we have identified key metabolic pathways and gene interactions that are influenced by individual drugs and their combinations. The combined drug networks, particularly those involving Cetuximab, Erlotinib, and Lapatinib, demonstrate how targeting multiple genes and pathways can potentially overcome resistance mechanisms commonly encountered in HER2+ breast cancer treatment. These findings underscore the need for a more integrated approach to drug development and precision oncology, where network-based analyses can guide the selection of drug combinations tailored to the unique molecular profiles of individual tumors. Future research should focus on validating these drug synergies in preclinical and clinical settings to optimize therapeutic strategies and improve outcomes for patients with HER2+ breast cancer, which may even open the door for treatments for other types of neoplasies.

Our results underscore the complexity of the ERBB2 signaling pathway and suggest new therapeutic opportunities by identifying potentially effective drugs and combinations in HER2-positive breast cancer. These findings need to be validated in additional studies, including clinical trials, to confirm their efficacy and safety. Our approaches may be incorporated as criteria during the evaluation of novel candidates for drug repurposing.

## Figures and Tables

**Figure 1 ijms-25-12840-f001:**
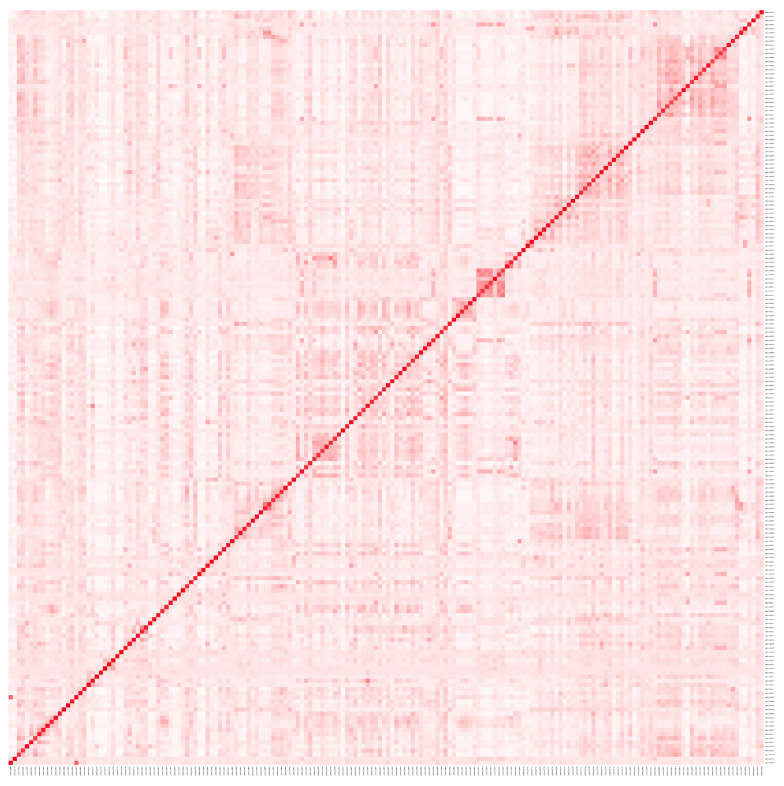
Heatmap displaying the Jaccard index values for the crossover between 181 metabolic pathways and the ERBB2 pathway. Each cell represents the percentage of genes shared between the ERBB2 pathway and another metabolic pathway, with color intensity indicating the degree of crossover: white indicates a zero percent or close-to-zero percent Jaccard index, reddish tones are proportional intermediate values, while red cells correspond to Jaccard values of one hundred percent (diagonal) or a closer value.

**Figure 2 ijms-25-12840-f002:**
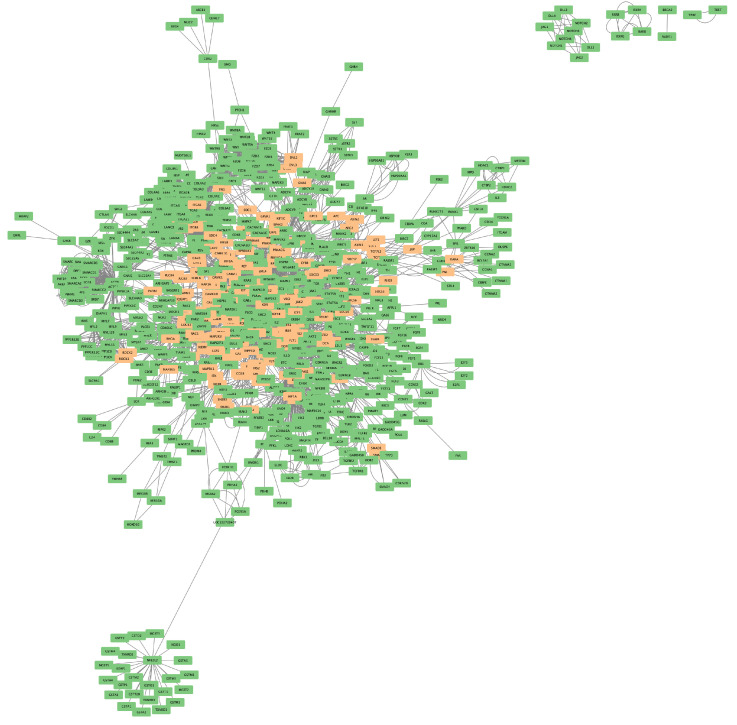
ERBB2 pathway-associated crosstalk network. Molecules depicted participate in one or more crosstalk events with the ERBB2 pathway (genes in the ERBB2 pathway depicted in orange).

**Figure 3 ijms-25-12840-f003:**
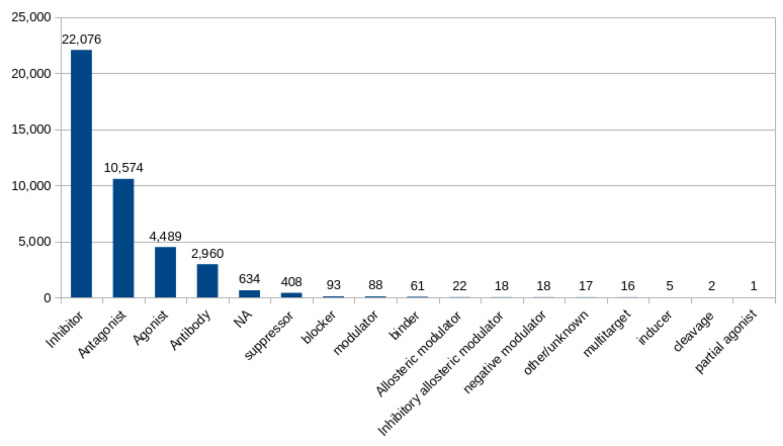
Bar graph showing the most common types of pharmacological interactions between drugs and the genes within the ERBB2 pathway network.

**Figure 4 ijms-25-12840-f004:**
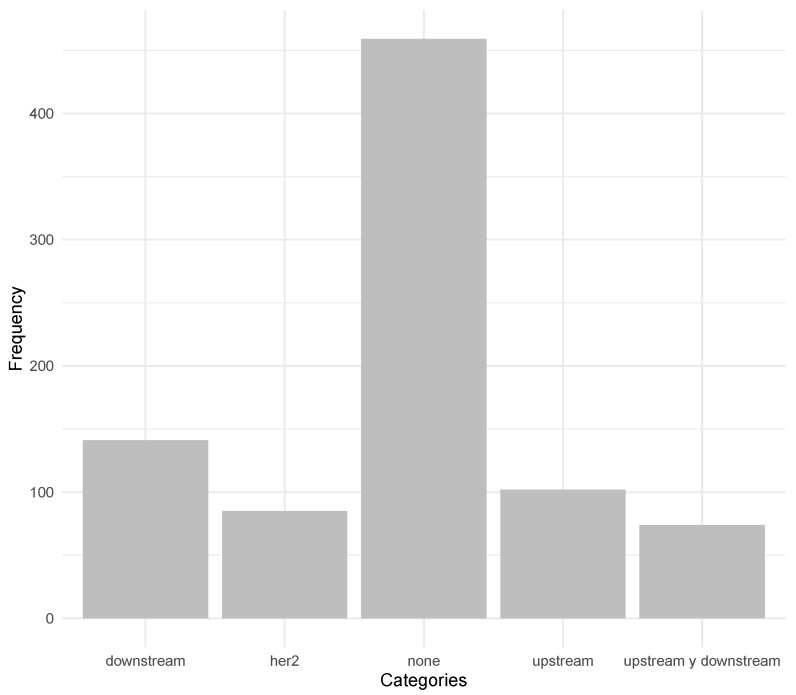
Bar chart categorizing the relationship of crosstalk genes in the 23 metabolic pathways with more than 20% crossover with the ERBB2 signaling pathway. Categories include downstream, upstream, ERBB2-specific, and variable (genes that can act both upstream and downstream depending on the crosstalking pathway).

**Figure 5 ijms-25-12840-f005:**
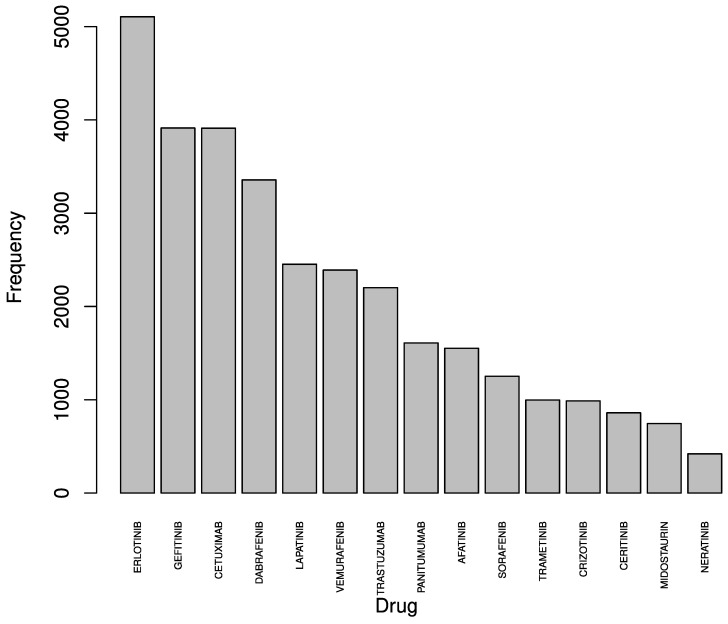
Bar graph showing the frequency of interactions between selected drugs and genes associated with the ERBB2 pathway. Drugs are ranked by their interaction count, highlighting Erlotinib, Gefitinib, Cetuximab, and Drabaneifb as having the highest number of interactions.

**Figure 6 ijms-25-12840-f006:**
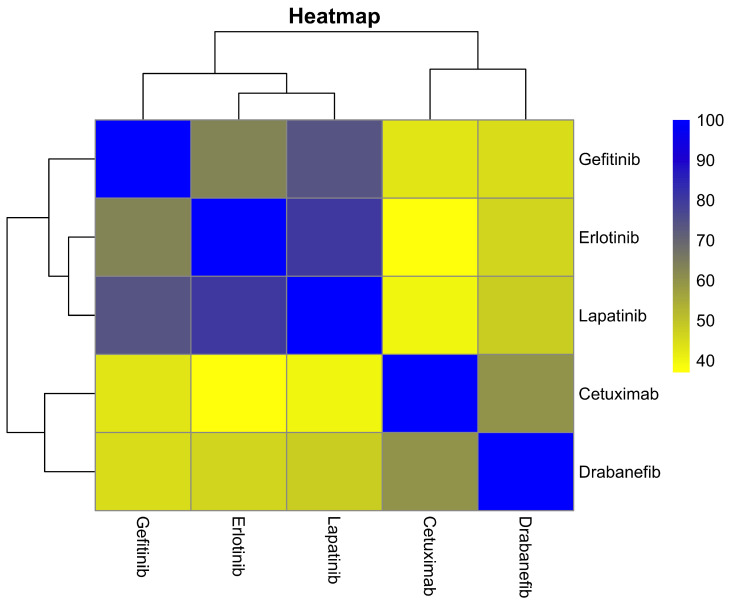
This heatmap depicts the Jaccard index for the top 10 drugs interacting with genes in the ERBB2 pathway, highlighting their target uniqueness. Darker colors indicate higher uniqueness, suggesting potential for drug synergy. Notably, Sorafenib, Dabrafenib, and Vemurafenib show distinct profiles, suitable for combination therapy in HER2+ breast cancer.

**Figure 7 ijms-25-12840-f007:**
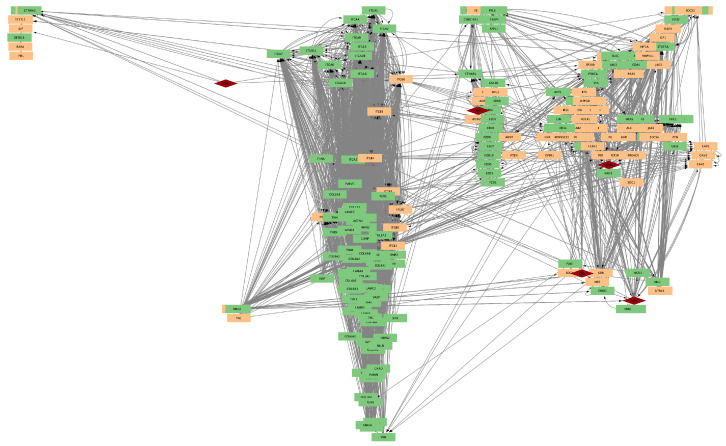
Coexpression network for genes affected by Erlotinib. Orange nodes belong to the HER2-pathway; red nodes are drug targets.

**Figure 8 ijms-25-12840-f008:**
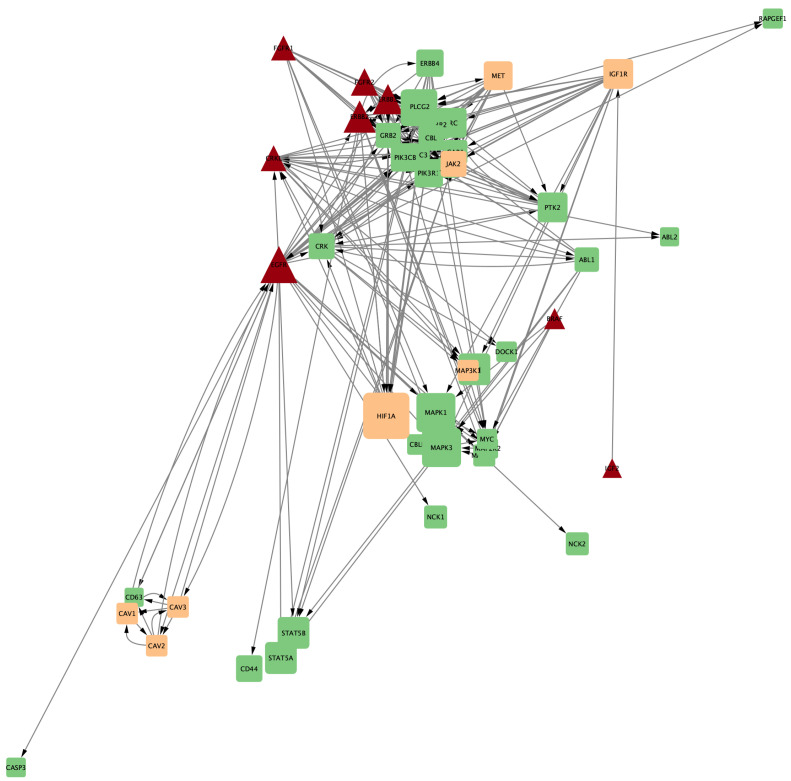
Coexpression network for genes affected by Gefitinib. Colors as in Figure 7.

**Figure 9 ijms-25-12840-f009:**
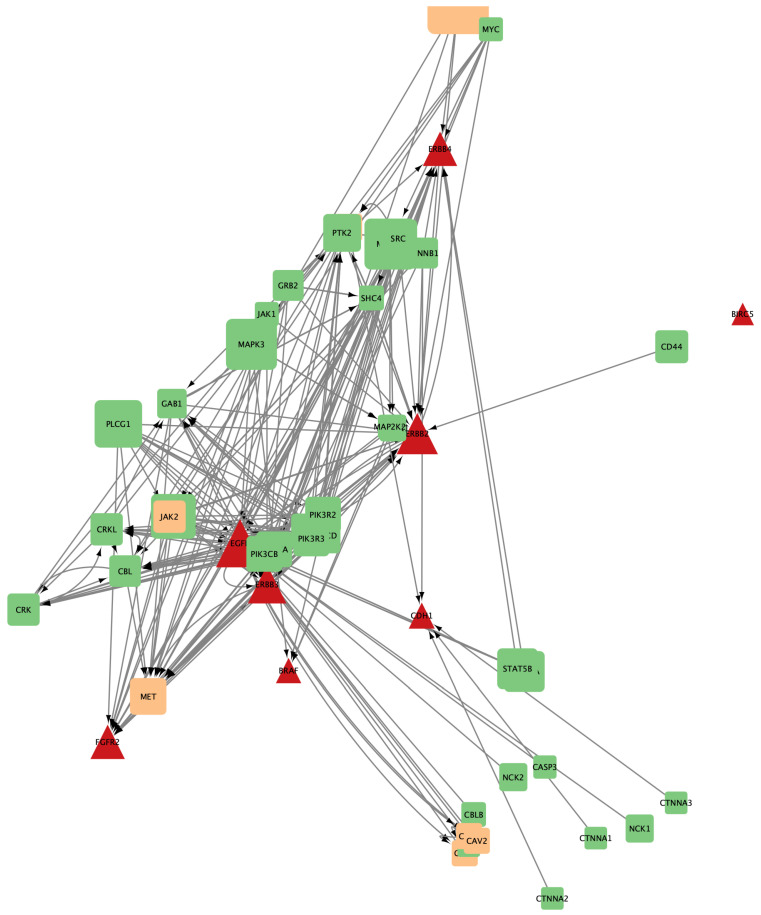
Coexpression network for genes affected by Lapatinib. Colors as in Figure 7.

**Figure 10 ijms-25-12840-f010:**
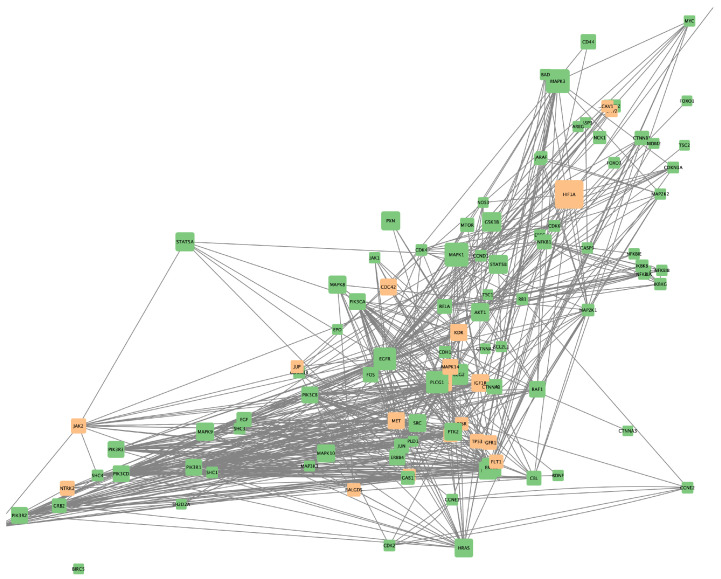
Coexpression network for genes affected by Cetuximab and Lapatinib. Colors as in Figure 7.

**Figure 11 ijms-25-12840-f011:**
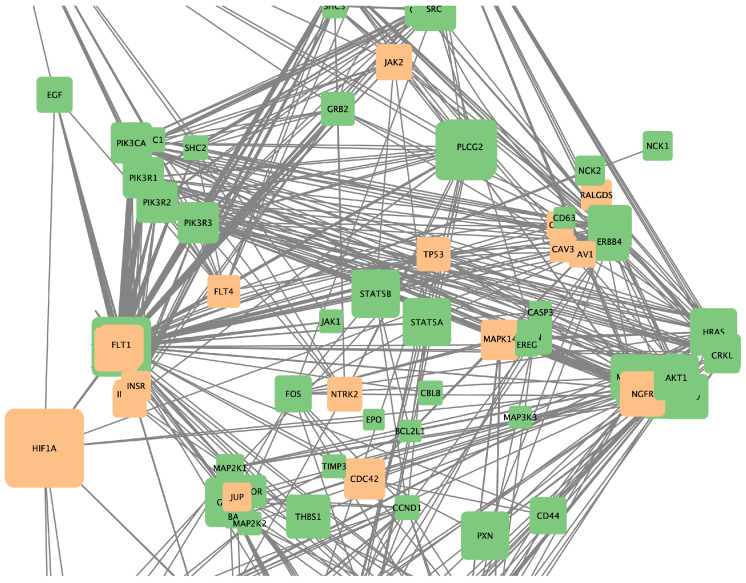
Coexpression network for genes affected by Erlotinib and Cetuximab. Colors as in Figure 7.

**Figure 12 ijms-25-12840-f012:**
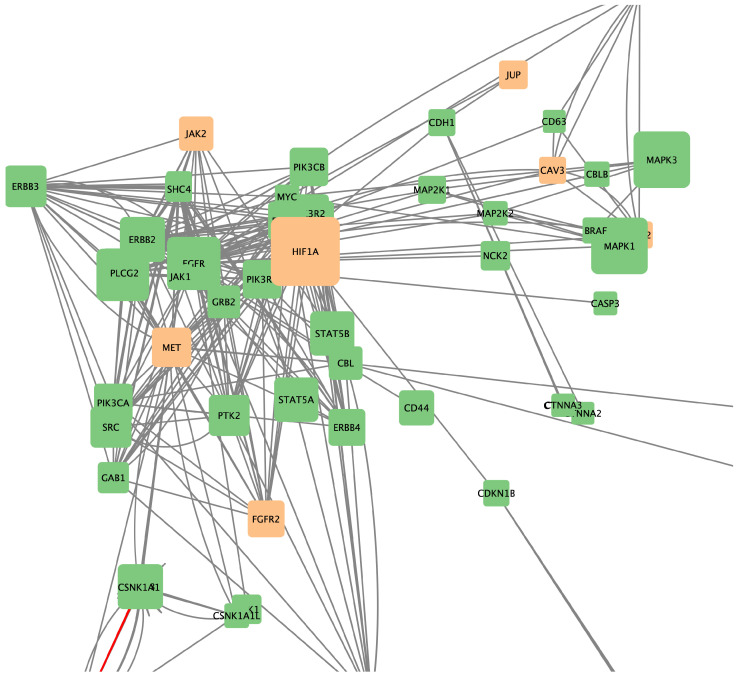
Coexpression network for genes affected by Erlotinib and Lapatinib. Colors as in Figure 7.

**Figure 13 ijms-25-12840-f013:**
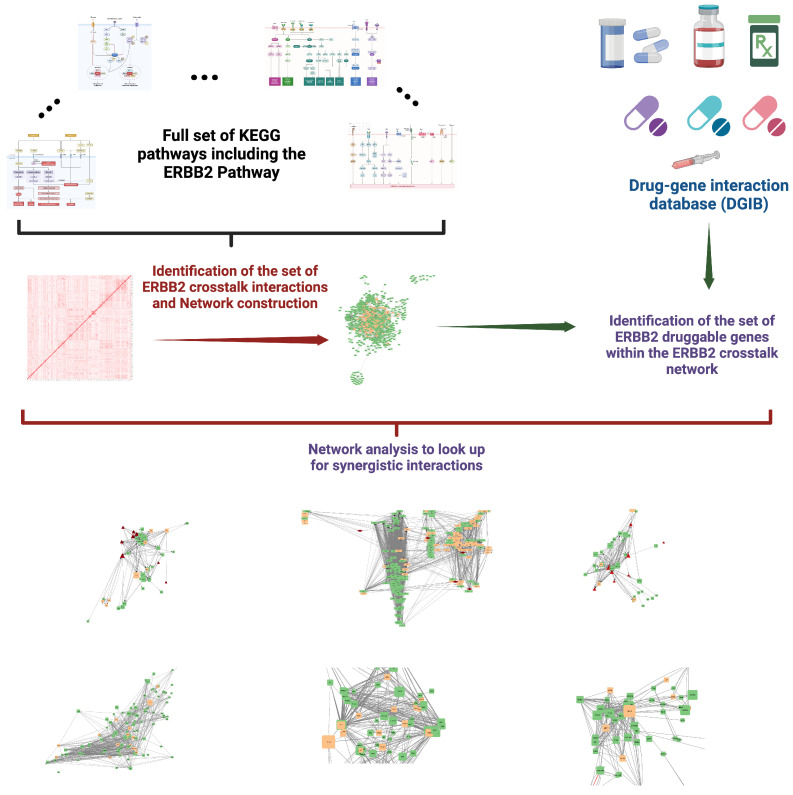
ERBB 2 crosstalk network and drug interactions workflow. Created in https://BioRender.com.

**Table 1 ijms-25-12840-t001:** Percentage of crossover with ERBB2 pathway.

Pathway	Percentage
EGFR Tyrosine Kinase Inhibitor Resistance	39%
Glioma	33%
Renal Cell Carcinoma	31%
Endocrine Resistance	29%
Colorectal Cancer	29%
PI3K Signaling Pathway	29%
Non-Small-Cell Lung Cancer	29%
mTOR Signaling Pathway	27%
Neurotrophin Signaling Pathway	26%
Prolactin Signaling Pathway	25%
AKT	25%
T-Cell Receptor Signaling Pathway	24%
Growth Hormone Synthesis, Secretion, and Action	24%
Choline Metabolism in Cancer	23%
VEGF Signaling Pathway	22%
Fc Epsilon RI Signaling Pathway	22%
PTEN Signaling Pathway	21%
Prostate Cancer	21%
Hepatocellular Carcinoma	20%
HIF-1 Signaling Pathway	20%
Focal Adhesion	20%
Proteoglycans in Cancer	20%
Gastric Cancer	20%

## Data Availability

Data is contained within the article.

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
