# Peer review of "Potential Drug Synergy Through the ERBB2 Pathway in HER2+ Breast Tumors"

_ijms, 2024, doi:10.3390/ijms252312840_

Round 1
Reviewer 1 Report
Comments and Suggestions for Authors
This manuscript explores the potential for drug synergy in treating HER2-positive breast cancer by analyzing the ERBB2 pathway and its crosstalk with other signaling pathways. Here's a breakdown of the strengths and weaknesses of the manuscript along with suggestions for improvement:
Strengths:
- Clear introduction: The introduction effectively establishes the background of HER2-positive breast cancer, limitations of current therapies, and the importance of pathway-based approaches.
- Detailed methodology: The methodology section clearly describes the methods used for network analysis, drug target identification, and network visualization.
- Exploration of crosstalk: The study investigates the crosstalk between the ERBB2 pathway and other metabolic pathways, highlighting the network's complexity.
- Identification of potential drugs: The identification of frequently interacting drugs like Erlotinib, Gefitinib, Cetuximab, and Lapatinib suggests potential therapeutic avenues.
- Visualization of drug-gene networks: The use of Cytoscape to visualize individual and combined drug-gene networks offers valuable insights into drug interactions.
Weaknesses:
- Limited discussion of existing literature: The discussion could be strengthened by incorporating relevant findings from existing studies on drug synergy in HER2-positive breast cancer.
- Lack of functional validation: The manuscript focuses on in silico analysis and lacks experimental validation of the identified drug synergies.
- Limited exploration of combination effects: While the study presents individual drug networks and some combined networks, a more in-depth analysis of specific drug combinations and their potential synergistic effects would be valuable.
- Absence of clinical data: The manuscript would benefit from including some discussion on the clinical relevance of the identified drugs and potential challenges in their application.
Suggestions for improvement:
- Include a more comprehensive review of existing literature on drug synergy in HER2-positive breast cancer, highlighting relevant findings and potential gaps addressed by this study.
- Discuss potential limitations of in silico analysis and emphasize the need for experimental validation of the identified drug synergies.
- Perform a more detailed analysis of specific drug combinations, exploring their potential synergistic effects on key signaling pathways and target genes.
- Include a section discussing the clinical relevance of the identified drugs, addressing factors like existing clinical trials, potential side effects, and limitations in their use.
- Consider revising the section on the HER2 pathway to provide a more concise overview, focusing on its role in HER2-positive breast cancer.
Additional comments:
- The figures are well-presented and informative.
- The categorization of genes involved in the ERBB2 signaling pathway (downstream, upstream, etc.) is a helpful addition.
Overall, this manuscript presents a valuable exploration of potential drug synergy in HER2-positive breast cancer. By addressing the suggested points and incorporating experimental validation, the study can be strengthened and provide a more comprehensive understanding of this promising therapeutic approach
Author Response
Please, find detailed responses in the attached PDF file.

Reviewer 2 Report
Comments and Suggestions for Authors
The article highlights the scope of drug synergies across the ERBB2 (HER2) pathway for breast cancers with HER2 overexpression. The ERBB2 pathway is instrumental in the development of aggressive tumor forms, often associated with less effective monoterapies owing to the phenomenon of tumor resistance.
Thus, I really think that herein lies the importance of this paper: the ERBB2 pathway is an important node for multiple-layer interactions that can be exploited for targeted therapy for breast cancer.
The paper, as stated, can be published if some big points are resolved:
-Introduction: this part of the article narrates a very expansive discussion regarding pathway interactions and treatment resistance in cancers. However, a brief description of the main research objectives, such as how drug combinations can enhance treatment and reduce resistance, would help the reader understand the overall direction of the study at the outset. It is suggested that the objectives of the study be clarified more directly and the value of the results be summarized.
-Methodology: Sometimes a complex network model can help elucidate a stepwise approach to the analysis in a flow chart. A flow chart showing some of the steps of the analysis would help us visualize how the analysis of the ERBB2 pathway blurred into the selection of a synergist. This would add clarity to the readers.
-Results: Although the results present a lot of data, it may be helpful to highlight the most important findings to focus the section. It may be useful to clearly distinguish between primary findings (eg, how the ERBB2 pathway is affected by certain drugs or how certain drug combinations increase therapeutic value) and secondary data.
-Heat maps and Jaccard index plots: these plots are useful, but an explanation of what each color gradation represents or what each interaction means would help to understand the significance of highly interacting pathways.
-Drug-gene interaction graphs: graphs showing drug-gene interactions in the ERBB2 network would be useful to be accompanied by a detailed explanation of what the nodes and connections represent to better clarify how these drugs affect pathways and their therapeutic value.
-Visualization of drug synergy networks: it is suggested to add a comment to each synergy network (eg cetuximab-lapatinib) to explain why the particular synergy is considered promising and the possible mechanism of action.
-Clinical application: it would be useful to further analyze how the selected drug combinations could be validated or have already been validated in a clinical setting. Discussion of preclinical level studies adds credibility to the approach and highlights necessary next steps before clinical implementation.
-Need for experimental validation: results are promising, but more detailed description of experimental validation methods (eg use of in vivo or in vitro models) is needed. Discussion of the need for validation increases the credibility of the proposed combination and provides a realistic path towards clinical application.
Author Response
Please find detailed responses in the attached PDF file.

Round 2
Reviewer 2 Report
Comments and Suggestions for Authors
The authors made all corrections. The revised version is completely agreeable to me, for this reason, I recommend the publication of the manuscript.